# Bioreactor Technologies for Enhanced Organoid Culture

**DOI:** 10.3390/ijms241411427

**Published:** 2023-07-13

**Authors:** Joseph P. Licata, Kyle H. Schwab, Yah-el Har-el, Jonathan A. Gerstenhaber, Peter I. Lelkes

**Affiliations:** 1Department of Bioengineering, College of Engineering, Temple University, Philadelphia, PA 19122, USA; joseph.licata@temple.edu (J.P.L.); kyle.schwab@temple.edu (K.H.S.); yahel@temple.edu (Y.-e.H.-e.); 2Neurobiology, Neurodegeneration & Repair Laboratory, National Eye Institute, National Institutes of Health, Bethesda, MD 20892, USA

**Keywords:** differentiation, biomimetic, stem cell, pluripotent, embryoid body, spheroid, stimulation, metabolism, simulated microgravity, 3D printing

## Abstract

An organoid is a 3D organization of cells that can recapitulate some of the structure and function of native tissue. Recent work has seen organoids gain prominence as a valuable model for studying tissue development, drug discovery, and potential clinical applications. The requirements for the successful culture of organoids in vitro differ significantly from those of traditional monolayer cell cultures. The generation and maturation of high-fidelity organoids entails developing and optimizing environmental conditions to provide the optimal cues for growth and 3D maturation, such as oxygenation, mechanical and fluidic activation, nutrition gradients, etc. To this end, we discuss the four main categories of bioreactors used for organoid culture: stirred bioreactors (SBR), microfluidic bioreactors (MFB), rotating wall vessels (RWV), and electrically stimulating (ES) bioreactors. We aim to lay out the state-of-the-art of both commercial and in-house developed bioreactor systems, their benefits to the culture of organoids derived from various cells and tissues, and the limitations of bioreactor technology, including sterilization, accessibility, and suitability and ease of use for long-term culture. Finally, we discuss future directions for improvements to existing bioreactor technology and how they may be used to enhance organoid culture for specific applications.

## 1. Introduction

Organoids have become increasingly useful in recent years by providing researchers with both complex models for studying organ development and robust disease models [1]. After decades of experience using “simple” 2D monolayer cell cultures for in vitro studies, it has become increasingly apparent that more complex tissue culture systems are needed to accurately recapitulate the structure and function of nascent and mature human tissues. In addition, organoids may help more realistically model the complexity seen in native tissues, including interactions with foreign microorganisms [2]. Recent advances in the use of organoids have led to their increasing application in disease modeling, cancer research, and the study of human development [3].

In this paper, the term “organoid” is defined as a 3D organization of cells that can recapitulate some of the structure and function of native tissue [4]. Nowadays, organoids are generally derived from stem cells, which may include embryonic stem cells (ESCs), induced pluripotent stem cells (iPSCs), or various tissue-specific adult stem cells such as mesenchymal stem cells (MSCs) or adipose tissue-derived stem cells (ASCs) [5]. Compared to monolayer cell cultures, organoids show appreciably more similarity to native tissues in terms of gene and protein expression, cell polarization, and macro- and micro-scale tissue organization. While we will discuss some of the bioreactors that have been used for other 3D tissue cultured constructs, such as embryoid bodies (suspension-grown aggregates of stem cells [6]) and spheroids (any spheroidal cluster of cells, regardless of structural complexity or function), our primary focus will be on how bioreactors may be used to enhance organoid research. As presented here, recent advances have largely focused on iPSC technology, which comes with many advantages, including the fact that it allows for patient-specific organoid development, streamlines eventual medical uses, and avoids any potential ethical issues surrounding the use of embryonic stem cells [7]. Furthermore, it should be noted that the optimization of organoid culture is focused on specific tissue generation, not the generation of whole organisms from stem cells. Recent developments have prompted discussion about the ethical issues surrounding the generation of embryos from human stem cells [8]. Research regarding artificial womb technology does exist [9], but these topics are outside the scope of this review.

Traditionally, the term “bioreactor” has been used to describe vessels designed to produce biological materials [10]. Commercial bioreactors are typically large tanks ranging from 1000 to 10,000 L in size, often containing a stirring or pumping mechanism for moving liquid into and out of the tank. These large-scale reactors are designed to facilitate bioprocesses, such as producing antibodies or other proteins from living cells and organisms. An essential feature of these devices is the ability to regulate environmental conditions, such as pH, dissolved gas concentration, flow/stirring rates, and nutrient levels in the system [11]. While many of these bioreactors were developed for the culture of bacteria, yeast, or other microorganisms, there have been significant advances in the large-scale culture of mammalian cells to produce biologically active proteins [12]. In addition, many biotechnology products that require post-translational protein modifications are often produced in large batch cultures of mammalian cells, such as Chinese hamster ovary (CHO) or baby hamster kidney (BHK) cells, as these cells can perform glycosylation and protein folding, which would not occur in bacterial proteins. As bioreactor applications for mammalian cell production have become common, additional strategies have been developed to better accommodate the requirements of mammalian cells, especially in regards to oxygen availability, the removal of metabolic waste products (e.g., CO_2_, lactic acid), and limiting excess shear stress [12].

A bioreactor, in the context of this paper, refers to any device used to enhance the 3D culture of cells and tissues in vitro. Specifically, we will discuss bioreactors and how they relate to organoid culture, though most of these technologies may also be used for other tissue culture constructs. Unlike commercial bioreactors used for large-scale protein production, most bioreactors currently used for organoid culture are small, using as little as a few milliliters of media [13]. Due to the high cost of cell culture media and supplements, most studies aim to limit bioreactor size. Such bioreactors can be used for targeted optimization at these scaled-down sizes, improving specific aspects of cell or tissue culture, such as the concentrations of oxygen and nutrients, or determining optimal flow rates and shear forces for a specific tissue. However, consideration is often given to scaling up a particular bioreactor design if needed [14]. Many potential organoid applications, such as high-throughput drug screening and cell-based therapeutics, require large numbers of cells or mature organoids. While consistent and relatively simple at low densities, monolayer culture can be challenging to scale [15]. The ability to conveniently scale up a specific bioreactor technology while keeping pilot studies small and cost-effective is highly desirable.

Dynamic bioreactors help to overcome many limitations of static culture, which is still the “gold standard” of tissue culture. Static 2D culture does not accurately recapitulate many aspects of an in vivo environment, including nutrient availability, gas exchange, substrate properties, waste removal, and more [16]. Using bioreactors for 3D culture allows researchers to mimic many of the environmental features not found in 2D cultures, such as increased fluid flow, gradients of signaling molecules and growth factors, and various types of stimulation [17]. To this end, we will discuss the most commonly used bioreactors for organoid culture, including stirred bioreactors, rotating wall vessels, microfluidic devices, and electrical stimulation bioreactors, as seen in Figure 1. This review will delve into the potential benefits and limitations of bioreactors for organoid culture.

## 2. Stirred Bioreactors

Stirred bioreactors (SBRs) for organoid culture have operating principles similar to that of the conventional, large-scale bioreactors used frequently for the commercial production of biomolecules. SBR cultures are characterized by their homogenous environment, simplistic monitoring, and straightforward control of key culture parameters, especially scalability. SBRs typically consist of a cylindrical culture vessel equipped with an impeller or agitator that can be driven directly or indirectly by a motor. Hydrodynamic forces are generated via two types of impellers: axial and radial flow impellers. The blades of axial flow impellers are pitched at an angle directing flow toward the base of the vessel, while the blades of radial flow impellers are perpendicular to the impeller shaft, generating a flow pattern directed toward the reactor’s walls, both shown in Figure 1A. Depending on the impeller configuration, rotational rate, and vessel geometry, fluid dynamic properties such as shear forces, nutrient exchange, and diffusion can be regulated to fit the needs of the culture [18]. Some examples of these may be found in Table 1. These impellers also perform various functions, including homogenization, heat and mass transfer, and aeration [19]. Significant efforts have been made to characterize the transport phenomena of these impellers, including oxygen transfer, nutrition mixing, and mechanical stimulation [20,21,22,23].

### 2.1. Improving Tissue Oxygenation

SBRs have been shown to provide numerous advantages for cell and organoid cultures, mainly stemming from improved mass transfer and enhanced oxygenation. Research efforts to improve differentiation using SBRs have indicated that media agitation can lead to increased cell survival and accelerated differentiation. Li et al. demonstrated that oxidative stress could be reduced during two-dimensional hematopoietic stem and progenitor cell culture by using SBRs. Their results showed that stirred conditions improve cell survival and induce differentiation at a faster rate compared to static conditions [24]. During three-dimensional tissue culture (i.e., organoid culture), the stationary diffusion of oxygen and nutrients is insufficient to meet the demands of developing tissues as they quickly increase in size and complexity [25]. Cellular spheroids exceeding 1 mm in diameter frequently exhibit a hypoxic, necrotic core encircled by a thin ring (~100 µm) of viable cells [23]. This issue becomes critical for tissues with high oxygen consumption rates, such as cerebral organoids. Lancaster et al. addressed this issue by demonstrating that SBRs improved oxygen availability and nutrient absorption, leading to the generation of complex cerebral organoids that were larger and more continuous than those grown in static conditions [26].

### 2.2. Achieving Proper Lineage Progression and 3D Spatial Organization

Progenitor cell proliferation, differentiation, and survival are vital for the formation of multilayered organoid tissue. However, this process poses challenges due to the distinct requirements of organoids during the early and late stratified stages. Complex tissue structures formed in static in vitro cultures often suffer from arrested development due to ischemic regions at the core of the structure caused by limited access to oxygen and nutrients [23]. Similar to removing the foundation of a building, these inner tissue layers collapse, potentially impairing the continued development and assembly of more metabolically active outer layers. The structural complexity of organoid tissue layers often depends on the extent of core cell survival. To improve organoid culture outcomes, it is necessary to use tools that enhance the diffusion of oxygen and nutrients to inner tissue layers. Efforts to achieve this goal using SBRs can be seen in the work performed by Qian et al., in which a custom spinning bioreactor designed in conjunction with 12-well plates is used to generate forebrain-specific cerebral organoids [27]. Compared to stationary and orbital shaker conditions, their SBR platform improved the diffusion of oxygen and nutrients, which facilitated the formation of large, continuous cortical structures. SBRs have also been implemented to advance lineage progression in cell cultures. Fluri et al. showed that the derivation of iPSCs from somatic cells cultured in spinner flasks removed the need for feeder-cell layers, serum, or costly tissue culture substrates (e.g., basement membrane matrix). Their approach, termed suspension culture-reprogrammed iPSCs (SiPSCs), was implemented by seeding inducible secondary mouse embryonic fibroblasts (MEFs) within treated spinner flasks and stirring at 65 RPM. The secondary MEFs showed increased annexin V staining during the first four days in suspension culture compared to their control (adherent). When cultured with doxycycline, a rapidly proliferating population of GFP-positive (cell having reprograming factors) and EdU-positive cells were developed, indicating actively reprogramed cells. Their results showed that the cells reprogrammed in these suspension cultures were comparable to those using conventional adherent monolayer conditions [28].

### 2.3. Types and Customizability

The most common type of SBR, the spinner flask, is a cylindrical glass container wherein individual cells or small cell aggregates are suspended to form the desired tissues. Considered the most minimal bioreactor, the spinner flask promotes the blending of oxygen and nutrients throughout the medium and decreases the concentration boundary layer on the surface of the nascent in vitro tissue constructs [18]. Modifications to the traditional spinner flask design have allowed for additional features such as the continuous perfusion of nutrient supply, waste removal, integration of sensor probes, pH control systems, and sampling ports [29]. SBRs can be easily scaled up compared to other culture systems, such as traditional static culture, rotating wall vessels, or microfluidic bioreactors [30]. For example, Qian et al. used a gear-operated stirring platform to assemble improved brain region-specific organoids [21]. Their SBR (SpinΩ) incorporated twelve impellers, each attached to a gear, driven by a single motor, with nearly all 3D-printed components fabricated in-house using 3D-printing.

Schneeberger et al. demonstrated the versatility that can be achieved with spinner flasks to cultivate matured hepatic organoids [31]. They focused on the large-scale production of hiPSC-derived liver organoids to address the unmet clinical need for liver transplantation alternatives. Their method involved expanding isolated single cells from liver biopsies and seeding them within spinner flasks set to 85 RPM, forming liver organoids between 11 and 14 days. To induce hepatic differentiation, organoids were transferred to a new spinner flask and differentiated for up to 12 days using a custom differentiation medium supplemented with 10% Matrigel. As seen in Figure 2, light microscopy images and hematoxylin and eosin (HE) and Villin-1 staining at day 5 indicated that the spinner flask organoids exhibited a more folded morphology than the control organoids with equal polarization. Culture in the spinner flask consistently led to the upregulation of hepatocyte markers (*CYP3A4*, *ALB*, *MRP2*) and downregulation of stem cell markers (*LRG5*) for all patient cells. Broad mRNA sequencing on day 12 of differentiation showed as much as four times upregulation of many genes associated with liver function and maturation (Figure 2C). The functionality of these hepatic organoids was demonstrated through enhanced MDR1-specific Rhodamine123 transport and increased glycogen storage in organoids cultured in a spinner flask versus static culture (Figure 2D). Overall, these results suggest that the large-scale expansion of patient-specific liver organoids using spinner flasks to produce semi-functional liver tissue is a promising approach, paving the way for more viable clinical applications.

### 2.4. Limitations of SBRs

Although the simplicity and ease of implementation make SBRs appealing, limitations such as high fluid shear and impeller instability may be concerning for shear-sensitive organoid cultures. Mammalian cells are shear-sensitive and can potentially be damaged by excessive hydrodynamic forces. If not appropriately designed, impeller geometry may introduce detrimental effects, inhibiting differentiation or destroying delicate structures such as cilia and microvilli. For example, Ovando-Roche et al. discovered that, although culturing retinal organoids within an SBR improved laminar stratification and increased the yield of cilia-bearing photoreceptors, there was a significant loss of fragile, late-stage outer segment-like structures [22]. The authors hypothesized that the shear stress induced by the SBR impaired the formation of these delicate structures. However, they noted that these effects could be limited by determining the optimal stirring rate that balances maximum cell yield and the preservation of fragile structures. Another limitation, particularly for custom-built and 3D-printed systems such as the SpinΩ (discussed earlier), is that increased complexity may lead to more technical difficulties for end users. However, this issue is constantly being considered by researchers, as evidenced by a later study creating the Spin∞, an iterative improvement on the SpinΩ system, which solved some issues regarding sterilization and reliability in long-term experiments [32].

**Table 1 ijms-24-11427-t001:** Selected studies using stirred bioreactors for organoid culture. hiPSC: human induced pluripotent stem cells; mESC: mouse embryonic stem cells; hESC: human embryonic stem cells; ASC: adipose-derived stem cell.

Reference	Cell Source	Goal	Bioreactor Set-Up	Outcomes
Lancaster et al. [25,26].	hESCs and hiPSC.	Generation of cerebral organoids using stirred bioreactors.	125 mL spinner flask.	Successful generation of organoids expressing markers for neurons and specific brain regions.
Qian et al. [21].	hiPSCs.	Region-specific brain organoids to model exposure to ZIKA virus.	Custom miniaturized spinning bioreactor (SpinΩ).	Successful generation of organoids specific to brain regions using a bioreactor.
Fluri et al. [28].	Fibroblast to hiPSCs.	Derivation and expansion of iPSCs from somatic cells.	Spinner flasks.	Culture in spinner flasks obviates need for feeder cells, serum, and solid substrate. iPSCs comparable to monolayer.
Zur Nieden et al. [30].	mESCs.	Pluripotent expansion and proliferation.	100 mL stirred flask.	hESCs in suspension maintain pluripotency and are easily scalable.
Ovando-Roche et al. [22].	hiPSCs.	Preservation of retina-like architecture and enhanced generation of photoreceptors .	100 mL stirred flask.	Improved laminar stratification and increase in photoreceptor yield.
Schneeberger et al. [31].	Patient-derived liver biopsies.	Generation of large-scale liver organoids.	100 mL spinner flask.	Culture in spinner flask increased the size and differentiation of the organoid.

## 3. Microfluidics

Microfluidic bioreactors (MFBs) are a promising platform for improving organoid cultures. MFBs combine the advantageous features seen in conventional 2D and 3D culture methods and other bioreactor platforms (e.g., induction of hydrodynamic forces) while offering additional benefits, including enhanced control over microenvironmental conditions and higher throughput. Examples of this can be found in Table 2. Generally, MFBs consist of one or more inlets, a series of small channels and chambers, and one or more outlets, as seen in Figure 1B. In addition, they are typically custom-designed with small architectural features ranging from millimeter to micrometer scale using micro-fabrication techniques such as photolithography-based molding [33]. These characteristics allow for a wide range of possibilities regarding culture device geometry, providing a dynamic platform for highly specialized applications.

### 3.1. Functions and Applications

The basic operating principles of MFBs involve transporting fluid volumes through channels and chambers via the use of pumps that are either externally attached to the device or integral to the system [34]. Inlet and output access ports interface between the culture area and the external environment. Most MFB systems use external pumps (e.g., syringes or peristaltic pumps) connected via tubing to the access ports, generating a perfusive flow. The primary purpose of this flow is to provide oxygen and nutrients and remove waste continuously. Moreover, secondary inlets (injection ports) can infuse additional supplements (e.g., chemicals, growth factors, enzymes). The laminar flow imposed by the channels’ geometries provides a mechanism to localize and regulate the delivery of these supplements, allowing for precise control over the spatial and temporal microenvironments surrounding the tissue constructs. Numerous types of sensor technologies have been implemented in MFBs to provide real-time experimental data. Optical sensing, spectroscopical approaches, and electrochemical assay methods have been used to monitor critical process variables, including oxygen, pH, CO_2_, glucose, and temperature [35,36,37]. For example, Zhang et al. fabricated a modular sensing MFB platform for liver and heart organoid culture that integrated physical, biochemical, and optical sensors [36]. This system consisted of a peristaltic pump, a media reservoir, a bioelectrochemical sensing module, a physical/chemical sensing module, a flow control breadboard module, a bubble trap, and organoid culture modules. These components are connected via tubing connected sequentially to form a complete circuit. Media is pumped from the reservoir through to the bubble trap, distributed by the flow control breadboard to the organoid modules, and then pumped through the sensing modules. The bioelectrochemical module employed a series of electrochemical microelectrode sensors that detect the electron transfer activity of redox probes upon antibody-antigen binding. To measure biomarkers, the authors considered albumin and glutathione S-transferase α as liver biomarkers and creatine kinase BM as a cardiac biomarker. These microelectrodes could also be cleaned and regenerated to detect other soluble biomarkers. The physical/chemical sensing module was integrated with optical oxygen and pH sensors and a temperature probe. In addition, integrated mini-microscopes at the bottom of the organoid culture modules obtained in situ morphology images in real-time. The module designed for organoid culture consisted of two hemi-chambers embedded between two layers (top layer: PDMS, bottom layer: PDMS/glass) sandwiched between two rigid PMMA supports. The liver organoids consisted of hepatic lobules formed from an aggregation of mask-pattern encapsulated human primary hepatocytes in gelatin methacryloyl (GelMA). Using a similar patterning technique, cardiac organoids were formed from hPSC-derived cardiomyocytes (CMs), seeded in parallel lines (50-μm spacing) of fibronectin-laden GelMA, and then covered with a layer of fibrin gel. After 3–4 days, the authors observed the alignment and stretching of the hPSC-CMs, which eventually formed cellular bridges between the parallel lines. Their results indicated that liver organoid modules could induce hepatocytes to aggregate into hepatic lobules (>85% cell viability at D5) while remaining functional, as indicated by the sustained secretion of albumin (measured by an electrochemical biosensor). The cardiac organoid module caused seeded cardiomyocytes to form cardiac organoids with cellular bridges. Using a mini-microscope, they observed strong and synchronized beating at a rate of ~60 beats/min, with uniform beating rates within a single batch of cells [36].

### 3.2. Advantages of Organoid Culture

As previously mentioned, the ability to precisely regulate microenvironment parameters using micro-scale geometries is a unique advantage offered by MFBs and is particularly significant considering the sensitivity of organoid cultures. The controlled way in which MFBs can deliver oxygen, nutrients, and other supplements is essential for improving organoid culture outcomes. As a result, using MFBs may increase the quality of organoid cultures, ensuring more uniform development, the directed differentiation of organoid cell types, and minimizing the unpredictability, heterogeneity, and lack of consistency that often result from conventional culture techniques or other bioreactor platforms [38,39]. Lastly, given that organoids are cultivated specifically to act as physiological models, the ability to simulate physiological conditions (e.g., generating gradients, applying temporal signaling molecules, and inducing or reducing mechanical force) provides a dynamic platform to simulate and investigate cell activity in response to specific stimuli [40,41,42].

### 3.3. Applications in Organoid Culture

Recently, Fu et al. demonstrated the impact that MFBs can have on the formation, culture, and analysis of multicellular spheroids [43]. In their work, an MFB was fabricated using an engraved acrylic mold to cast (PDMS) channeled microchambers. This paper highlighted the ability to use unusual lithography to generate complex geometries. Briefly, photolithography was used to generate a photomask on which a photoreactive polyethylene glycol precursor was polymerized to form an array of U-shaped structures within the microchamber, leading to the fabrication of the MFB. Balb/c 3T3 fibroblasts and human hepatoma cell line HepG2 suspensions were loaded into the MFB and trapped within its U-shaped structures. The trapped cells were cultured under continuous media perfusion using a peristaltic pump at a flow rate of 8 μL/min to form multicellular spheroids. Once the spheroids were formed, the flow rate was reduced to 1.5 μL/min for long-term culturing. Their results indicated that many homogenous multicellular spheroids could be generated in this MFB using fluid flow and unit gravity sedimentation. The material and geometry of the hydrogel employed allowed for an efficient exchange of nutrients and waste while shielding the organoids from perfusion-induced shear stress.

Wang et al. introduced another approach using MFBs equipped with a perfusable micropillar array in which hiPSC-derived liver organoids could be maintained in long-term culture [44]. Their MFB was fabricated using soft lithography and included patterned micropillars and inlet and outlet ports, as seen in Figure 3. Analysis of the liver organoid growth involved examining the size distribution using the average diameters measured from day 0 to day 30, as shown in Figure 2D. These data indicated a positive correlation between the progression of differentiation and the average size of the organoids. Furthermore, the liver organoids demonstrated enhanced cell viability when subjected to perfused culture conditions on the chip in comparison to the static cultures seen in Figure 3E. This was determined through IHC staining for active caspase 3 on day 30. Quantitative analysis revealed a significant amount of cell death within the organoids in the static cultures, with approximately 12% of the cells exhibiting caspase 3 positive staining (see Figure 3F). By contrast, the use of perfusion culture in the MFB system can improve cell viability, as indicated by the significant reduction in caspase staining in the organoids grown in a dynamic environment.

Additional evidence of the advantages of using MFBs to cultivate organoids can be found in the research performed by Achberger et al. [45]. They fabricated an MFB, deemed a Retina-on-a-chip (RoC), to culture retinal organoids (RO) and retinal pigmented epithelium (RPE) in a manner that more closely resembles physiological conditions (see Figure 4). In this work, a porous polyethylene terephthalate (PET) membrane was placed between two photo-patterned PDMS wafers, the bottom one containing a media perfusion channel and the top one containing wells to house the RPE and RO. First, the PET membrane at the bottom of the well was coated with laminin. Subsequently, RPE were seeded on top and allowed to adhere to the membrane. Next, ROs were placed onto a membrane confluent with RPE, and a hyaluronic acid-based hydrogel was added to each well. Finally, media was perfused through the MFB at 20 μL/min. Their results showed roughly three times greater outer segment formation using the MFB platform compared to conventional static culture conditions. See Figure 4 for additional details.

### 3.4. Limitations of MFBs with Respect to Organoid Cultures

Despite the advantages offered by MFBs, multiple limitations and associated challenges hinder their widespread use. By design, intricate structures designed to control fluid flow and generate optimal conditions are required, making MFB fabrication difficult and time-consuming [46]. Furthermore, some of the materials used in the bioreactor fabrication process may not be fully biocompatible, potentially degrading over time and tending to carry out the adsorption and adhesion of unwanted biomolecules. These issues can alter cell behavior, generate hazardous byproducts, and reduce fluid flow, which can be detrimental to the success of organoid culture [36]. The most prominent limitation of MFBs is related to their base component fabrication. For example, soft lithography molding involves patterning 2D micro-features on PDMS and often requires photoresist mold manufacturing. The industry standard method involves forming molds by utilizing photoresist resin (epoxy) on wafers (silicon, quartz) to achieve a high feature resolution. This process requires a clean room, preprocessing equipment (photomask fabrication, wafer preparation), specialized projection equipment (UV, X-ray, e-beam), and postprocessing (development, cleaning, surface treatment) [47]. Although some companies offer this fabrication, cost-per-unit mold prices are high and vary broadly based on pattern complexity and the materials used. In addition, designing MFBs is often an iterative process that requires numerous modifications to achieve the desired results. As such, utilizing soft lithography methods is often non-sustainable.

Researchers looking to incorporate MFBs into their workflow may experience barriers to access due to the specialized knowledge and abilities required to fabricate or operate most custom-made or commercially available MFB platforms. Similarly, scaling up these systems for larger-scale experimentation remains challenging due to the necessity of maintaining the precise control of fluid dynamics and ensuring uniformity across many devices [48]. Furthermore, long-term experiments may become contaminated due to prolonged contact with the environment, mainly if the device is not adequately sealed or sterilized [49]. As such, time-consuming procedures for cleaning and sterilizing small, complicated components are necessary and present additional maintenance problems [50].

Successful, long-term organoid culture requires constant monitoring of the culture conditions and assessment (periodic or continual) of the development and maturation of the organoids. However, the complexity of MFBs often impairs the ability to quickly and repeatedly ascertain the status of the cultures, requiring time-consuming intervention, which often includes disconnection from a perfusion system and removal of cultured tissue from the device. The alternative to this approach would involve additional equipment or techniques such as specialized sensors or in situ microscopy; however, these are often expensive and/or difficult to integrate [51]. In addition, the overall cost can be concerning, as the fabrication of custom-made devices or acquisition of commercial MFBs can be expensive. This is especially concerning if the bioreactors require continuous design modifications, proprietary materials such as low-contamination connectors, or expensive consumables such as filters, reagents, and culture media [51].

**Table 2 ijms-24-11427-t002:** Selected studies utilizing microfluidic bioreactors for organoid culture. hiPSC: human induced pluripotent stem cells; BMSC: Bone marrow-derived mesenchymal stem/stromal cells.

Reference	Cell Source	Goal	Bioreactor Setup	Outcomes
Zhang et al. [36].	HepG2, hiPSCs.	Integration of multiple sensors for automated and continuous in situ monitoring.	Layered PDMS microbioreactor integrated with microelectrode array.	Real-time monitoring of microenvironmental parameters, soluble biomarkers, and organoid morphology.
Toh et al. [42].	HepG2, MCF7, BMSCs.	Improvement of cell–cell and cell–matrix interaction via perfusion-seeding and entrapment.	Perfused PDMS substrate equipped with microchannel and micropillar array.	Preservation of 3D cytoarchitecture, differentiation competence, and cell-specific function. Direct microscopy monitoring.
Fu et al. [43].	HepG2 and Balb/c 3T3 fibroblast cells.	Enhanced formation of multicellular spheroids using a microfluidic Bioreactor.	Two-port PDMS perfusion bioreactor with U-shaped hydrogel structures.	Successful trapping of cells and spheroid formation in non-adherent hydrogel structures.
Wang et al. [44].	hiPSCs.	Creation of a chip-based liver model.	Perfused PDMS micropillar array.	Better tissue functionality from a perfused chip system than static culture.
Achberger et al. [45].	hiPSCs.	Biomimetic model of subretinal space vascularization and cell interaction.	Hydrogel-laden PDMS chamber with PET membrane on a glass slide.	Vascular-like perfusion improved the interaction of the photoreceptor, and RPE enhanced the outer segment-like structures.

## 4. Rotating Wall Vessels

Rotating Wall Vessels (RWVs) have their roots in the clinostat, a device that can rotate an object on an axis (or multiple), and which was developed in the early 1800s for the culture of plants [52]. RWVs, first developed by NASA [53], work similarly to the clinostat as they use the rotation of a circular or cylindrical vessel to create low-shear mixing and simulated microgravity within the vessel [54]. RWVs generally come in two types, either the slow-turning lateral vessel (STLV, cylindrical) or the high aspect ratio vessel (HARV, circular), both shown in Figure 1C [55]. It has been shown that, in either type of RWV, the synchronous rotation of the cell culture (cells/cell assemblies and culture fluid) with that of the entire vessel allows for the “solid-body” rotation of the liquid within the vessel, which leads to the gentle, low-shear mixing characteristic of these bioreactors while also creating the continuous tumbling motion that simulates microgravity [55]. The STLV and HARV mainly differ due to their overall shape and aeration source, which is a central cylindrical oxygenator in the STLV and a gas-permeable membrane on a wall of the HARV [56]. They also differ in that the STLV can accommodate much larger volumes, often including the continuous perfusion of media into/out of the bioreactor. Comparatively, HARVs provide more oxygenation and thus are the device of choice for organoid culture, while STLVs are frequently used in cultures that require precisely controlled oxygen levels (e.g., hypoxia) [54].

### 4.1. Uses in Organoid Culture

RWVs have numerous benefits for cell culture in general and for organoid culture specifically (see Table 3 for examples). As one of the most critical aspects of RWVs, the significant enhancements in mixing in a low-shear environment results in enhanced cell performance [57]. For example, by culturing cells in an RWV, compared to spinner flask culture, recombinant protein production increased seven-fold in a human cell line [58]. While the traditional spinner flask also enhanced nutrient availability to the cells, the high-shear environment in those flasks caused enough damage to the cells to reduce their activity. Regarding the multiple variations in RWV design, through modeling and experimentation, it has been shown that fluid shear stress is significantly lower in RWVs than in other types of mixing vessels [59]. This low-shear mixing in the RWVs is essential for organoid culture because, as spheroids grow larger, they tend to develop a necrotic core due to a lack of nutrient availability [60]. Enhanced media mixing helps to mitigate this, but only if it can be achieved without introducing adverse effects from the high shear. One of the limitations of this lower shear is that it might limit the cells’ mass transport and, consequently, nutrient availability.

RWVs serve multiple purposes specific to organoid culture. The first of these is spheroid formation, generally performed by culturing cells at high density within the RWV and allowing them to self-aggregate [6]. The formation of spheroids in RWVs has been well documented, showing initial aggregation and growth over time [61]. Another use involves the expansion of pluripotent stem cells in vitro. One challenge is that conventional 2D stem cell culture often requires expensive media and an expensive matrix to coat the tissue culture plastic to increase cell adhesion [62]. RWVs may be an effective tool for growing large numbers of stem cells in suspension while maintaining pluripotency, as suspension culture eliminates the need for expensive matrix proteins as cell culture substrates [30]. Rogers et al. cultured human mesenchymal stem cells (hMSCs) in a 55 mL commercial RWV system using inexpensive gelatin-based microcarriers to aid in embryoid body culture [63]. They demonstrated that, when using a 7.5% gelatin methacryloyl (GelMA) microcarrier in an RWV, the population of hMSC cells could be expanded 16-fold within eight days. The cells maintained their adipogenic and osteogenic differentiation capabilities, as well as their immunomodulatory potential. Thus, RWVs have successfully provided a method for the scalable expansion of stem cells that does not require expensive matrix proteins.

The differentiation of organoids into specific tissue types using RWVs has been extensively investigated. For example, Wilkinson et al. used a small (4 mL) HARV-type RWV to develop lung tissue organoids by seeding collagen-coated alginate beads and iPSC-derived lung fibroblasts or fetal lung fibroblasts into an RWV and allowing the organoids to self-assemble into a complex structure [64]. In this case, the use of the RWV helped form a lung-tissue model for disease modeling and drug screening [64]. In another study, DiStefano et al. attempted to grow retinal organoids in RWV culture [65]. After ten days of initial 3D spheroid formation under static suspension culture conditions, some of the nascent organoids were transferred into the bioreactor, while the control group was left in static suspension culture. The organoids in the RWV grew significantly faster and matured earlier than those in static culture, as seen in Figure 5. Specifically, the RWV organoids showed a level of maturation at day 25 that was not seen in the control group until day 32, shaving one week off the necessary culture time. RNA-seq transcriptome analysis was used to confirm that the pattern of gene expression in the organoids maintained in the RWV matched that of an in vivo retina at an earlier timepoint. The authors hypothesized that enhanced oxygen, nutrient, and waste exchange led to faster growth and maturation [65].

### 4.2. Simulation of Microgravity

Originally, RWV bioreactors were developed by NASA to serve as a cell/tissue culture venue that might simulate microgravity on Earth. Though the previous discussions in this review have focused on the benefits for organoid culture in general, it is important to remember that, when used properly, the RWV will also yield an analog of certain aspects of reduced gravity (~10^−3^× *g*), also known as “simulated microgravity” (SMG). For example, neuronal/neuroendocrine PC12 cells (derived from a tumor of rat adrenal) have been extensively studied in SMG conditions [66]. In a further study on neuroendocrine interactions [67], upon seeding single cells into the RWV, the cells aggregated, forming a neuroendocrine organoid, which were cultured for up to 30 days. After one day in SMG in the RWV, multiple pathways, including the ERK, p38, and jnk signaling pathways and the adrenergic enzyme PNMT, were upregulated compared to static conditions (VueLife^®^ bags). After 30 days of culture in various conditions, the organoids were implanted subcutaneously in mice. Significant vascularization occurred only in the organoids that had been cultured in the RWV. This study tried to account for the effect of shear stress and mixing versus that of microgravity. Notably, the PC12 cultured in a stirred bioreactor at the same shear stress as the RWV did aggregate; however, they did not show the differentiation and organization seen in RWV-cultured organoids. These results suggest that the effect of microgravity may act independently of the effects of fluid mixing. This observation is reminiscent of prior studies in the Lelkes lab with neuroendocrine organoids, suggesting an upregulation of angiogenic factors (e.g., VEGF) in the organoids grown in the dynamic environment of RWV bioreactors (see Figure 6). In this study, significantly more vascularization was seen in organoids grown in a HARV compared to those in static suspension culture.

### 4.3. Limitations

While RWVs are potentially advantageous for organoid culture, there are some limitations, the first of which is the relatively high user competency required. Cell seeding and media exchanges require particular care to avoid introducing bubbles into the vessel. Computational modeling has shown that the presence of an air bubble within an RWV can give rise to localized regions of substantially elevated shear forces relative to the surrounding environment. This is due to the bubble’s tendency to remain located at the top of the RWV while the liquid continues to rotate around it [68]. Because of their detrimental effects, removing any bubbles from the device is essential. One way to remove bubbles is to use the recently described “bubble isolating” HARV-type RWV (see Figure 7) [68], in which bubbles are continually sequestered in a thin exterior concentric channel running along ∼80% of the total perimeter of the HARV and connected to the main body by a thin entrance. This design may be custom-built from low-cost materials. Further, a variation of this design incorporates perfusion into a HARV, therefore using the bubble-catching design [69]. By continuously blending media from a more extensive reservoir into the HARV, this design reduces the need for frequent media changes, simplifying operation for end users. This is particularly important for projects that may need to be scaled up in the future.

Another challenge for the user is determining the correct speed for maintaining the balance between the solid-body rotation and the continuous freefall of organoids within an RWV. The exact speed required for optimal low-shear rotation will vary based on the size/density of the aggregates/organoids and their sedimentation coefficient within the vessel. For uniform cultures, in which the size of the organoids increases with time, the rotational speed must be increased over time to maintain the above-mentioned balance. For example, DiStefano et al. showed that they needed to increase their RWV rotational speed from 20 to 27 RPM over two weeks to maintain solid-body rotation while the retinal organoids grew [65]. However, for heterogeneous organoid cultures in which the sizes of the organoids in the RWV vary significantly in size and density, it may be impossible to find a rotational speed that is appropriate for all particles.

The final limitation of RWVs is inherent in their design, namely, the simulation of microgravity. While RWVs can and have been used in modeling the effect of microgravity in various tissues (as discussed previously), users must consider that when microgravity simulation is not the intended focus of a study, the SMG environment may actually have detrimental effects on the cultures. Some of the documented effects of SMG include altered calcium handling in cardiac cells [70], a reduction in chondrogenesis [71], and interference with cellular differentiation pathways, such as the downregulation of osteogenic markers ALPL and OMD during osteogenic differentiation performed in SMG [72]. Therefore, proper controls must be designed to discern whether any changes to cells cultured in an RWV are due to enhanced mixing, low shear stress, or microgravity simulation. However, as SMG effects may be detrimental to specific cell and tissue types, not all organoids may be appropriate for culture in an RWV, particularly musculoskeletal organoids [73], unless the intention is to study the effects of SMG on that tissue.

**Table 3 ijms-24-11427-t003:** Selected studies utilizing rotating wall vessel bioreactors for organoid culture. hiPSC: human induced pluripotent stem cells; mESC: mouse embryonic stem cells; hESC: human embryonic stem cells.

Reference	Cell Source	Goal	Bioreactor Setup	Outcomes
Rogers et al. [63].	Human mesenchymal stem cells.	Embryoid Body cell expansion.	10 mL HARV from Synthecon, GelMA microcarriers.	Cells maintain pluripotency and differentiation capacity as well as or better than monolayer culture.
Botta et al. [61].	PC12 cells.	Spheroid formation.	50 mL HARV from Synthecon, microcarrier beads.	Real-time imaging shows an increase in aggregate size over time.
Wilkinson et al. [64].	iPSC-derived lung fibroblasts.	Lung organoid generation.	4 mL HARV from Synthecon, alginate beads.	Lung organoids generated in a scalable manner appropriate for high-throughput screening.
DiStefano et al. [65].	mESCs.	Improved retinal organoid generation.	50 mL HARV from Synthecon.	Faster growth and maturation in RWV compared to static, as shown by fluorescence imaging and RNA-seq.
Takahashi et al. [74].	Human iPSCs.	Intestinal organoid.	JTEC CellPet 3-D Ips [75].	More efficient, robust, and scalable organoid generation.
Papadaki et al. [76].	Rat primary cardiomyocytes.	Cardiac Organoid.	100 mL HARV from Synthecon.	HARV-cultured organoids have improved contraction and cardiac gene expression.
Lelkes et al. [67].	PC12 cells.	Neuroendocrine organoid culture.	35 mL HARV from Synthecon.	Increase in neuroendocrine expression over neuronal expression and increased vascularization with RWV culture.
Phelan et al. [68].	A549 human lung adenocarcinoma cells.	Cancer organoid.	Custom 10 mL air bubble-isolating RWV.	RWV without bubbles led to significant increases in organoid size.

## 5. Electrical Stimulation Bioreactors

Electrical Stimulation (ES) bioreactors are defined here as any bioreactor capable of providing some type of ES to cells and organoids cultured within the device. Although there are several approaches to providing ES to cells, such as capacitive coupling and inductive stimulation, the most common method is direct stimulation. This involves inserting electrodes directly into the culture media on either side of the cells of interest, as seen in Figure 1D [77]. Using this method, electricity can pass directly through cells and organoids/tissues. Some tissues are considered “electrically excitable”, containing cells that can respond to or generate electrical signals [78]. Numerous tissues cultured in vitro have benefitted from external electrical stimulation as a tool for modulating growth, differentiation, and maturation [77]. Some examples of these may be found in Table 4. Foremost among those excitable tissues are neural tissue (including retinal and other peripheral nervous tissue) and muscles (including cardiac tissue). Endogenous electric fields within the heart contribute to cell differentiation, orientation, proliferation and the upregulation of various signaling factors [79]. At an organ/tissue level, electric fields contribute to patterning tissues and cells [80].

ES bioreactor systems vary in terms of design and the materials used, but most contain the same essential components, namely, electrodes and some systems dedicated to providing the electricity for the stimulation. In most cases, electrical stimulation is not delivered continuously; instead, it is delivered using electrical pulses of specific frequency, duration, and amplitude [81]. While some ES bioreactor designs are basic, i.e., essentially consisting of two electrodes on either side of the cells/organoids of interest, some designs can be more complex. Multi-electrode arrays (MEAs) have been used extensively for both the stimulation and recording of electrical signals from cell monolayers [82], but variations have recently been developed for providing ES to brain organoids [83]. These recent MEA developments for organoid use have centered around creating a more complex 3D MEA that is able to stimulate specific areas of the organoid without necessarily stimulating the entire organoid.

### 5.1. Organoid Electrical Stimulation

The application of ES aids in the differentiation and maturation of organoids of excitable tissues (see Table 4). For example, stem cell-derived cardiomyocytes cannot fully mature under standard in vitro culture conditions as they lack the enhanced morphology, electrophysiology, calcium handling, contractility, and metabolism seen in mature cardiomyocytes in vivo [84]. Electrical stimulation is one method that has been used to enhance cardiomyocyte maturation in 3D organoids. Using cardiac embryoid bodies/organoids, one study used a custom bioreactor consisting of carbon rods embedded into the side of PDMS (polydimethylsiloxane) wells (see Figure 8A) [85]. The organoids were subjected to electrical stimulation on days 23–30 of differentiation using a 5 V/cm amplitude and 2 ms pulses at a frequency of either 0.5, 1, or 2 Hz. Overall, ES led to an increase in cardiac gene expression and cell–cell connection, as can be seen in the data summarized in Figure 8. In addition, as the frequency increased, the authors reported an adaptation to beating at higher frequencies. For 2 Hz, after stopping ES, the organoids continued spontaneous contractions at or close to 2 Hz, whereas non-stimulated organoids exhibited a slower spontaneous contraction. Furthermore, conditioning organoids at 2 Hz allowed them to survive 3 Hz stimulation, whereas non-conditioned organoids saw reduced survival at 3 Hz ES. In another study, increasing the stimulation frequency from 1 Hz to 2 Hz led to increases in cardiac troponin expression and sarcomere organization in cardiac organoids formed from hiPSC-derived cardiomyocytes [86].

While these studies sought to determine whether ES would enhance the maturation of cardiac organoids, ES has also been used to induce early-stage cardiac differentiation. For example, Ma et al. used an IonOptix C-Pace commercial stimulation system [87] to initiate ES after seven days of organoid formation and observed beating cardiac organoids in as little as two days after ES initiation (compared to 7 days without ES) [88]. After 2 weeks of culture with ES, the spontaneous flux of calcium ions was increased compared to the control group, as observed using an engineered GCaMP calcium indicator under the control of a cardiac troponin reporter. With ES, the proportion of cells expressing cardiac troponin and Nkx2-5 cells was increased, and many genes for ion channels and cardiac transcription were upregulated, as shown via the use of RNA-seq analysis. The authors concluded that electrical stimulation alone, i.e., without the use of any external chemical signaling, could improve early cardiac differentiation.

ES bioreactors have also been used to enhance the culture of neural organoids. Aiming to stimulate cortical brain organoids cultured from primary mouse cortical neurons, Zhang et al. designed a system of interdigitated electrodes, as seen in Figure 9 [89]. This system was used to apply ES to both wild-type(WT) and neuregulin 1 knock-out (NRG1-KO) neural organoids. A comparison was made between the WT and NRG1-KO lines with and without electrical stimulation. Exposure to ES led to an improvement in the number and length of neurites and increased the expression of markers for synaptogenesis, including NCAM and PSD95, in the NRG1-KO organoids, showing the ability of ES to rescue the phenotype and promote the development of dysfunctional neurons (see Figure 9 for more details). Sefton et al. studied the effects of electrical stimulation on the formation and differentiation of neurospheres made from mouse neural precursor cells [90]. Using electrodes made from platinum wires, they found that in vitro stimulation increased both cell survival and the size of neurospheres formed. Furthermore, electrical stimulation led to an increased fraction of cells displaying neuronal and oligodendrocyte markers, while the fraction of cells exhibiting astrocytic markers declined. This suggests that electrical stimulation may influence the fate determination of neural cells in their early stages.

### 5.2. Commercial vs. Custom-Made Systems

Multiple commercial devices have been used for the electrical stimulation of cells and organoids; to the best of our knowledge, the most common system is the Culture Pacing System (C-Pace) by IonOptix [87]. This system includes a series of custom tissue culture plates utilizing carbon electrodes that extend from the plate lids down to the culture surface. This also includes the signal generator for creating the ES pulses, which have been used to provide ES to the custom electrodes. Specifically, the IonOptix system has been used in numerous studies using ES culture organoids [86,88,91,92,93] due to its ease of use as an out-of-the-box solution for providing ES to cell/organoid cultures. However, some researchers prefer to develop fully custom ES setups, some of which were mentioned previously in this paper. Reasons for this may include the ability to further customize the system, including the use of different electrode materials [94], different culture vessels [95], and more specific control over the stimulation parameters [96]. In addition, custom-designed systems may allow for the incorporation of features not found in commercial systems, such as the addition of media perfusion [97] or the combination of electrical and mechanical stimulation [98]. Another factor is cost, as many custom-made systems can be made in-house for relatively little money. As discussed above, the most basic ES systems mainly involve only electrodes being inserted into tissue culture plates, which may be accomplished for a very low cost.

**Table 4 ijms-24-11427-t004:** Selected studies utilizing electrically stimulated bioreactors for organoid culture. hiPSC: human induced pluripotent stem cells; mESC: mouse embryonic stem cells; hESC: human embryonic stem cells.

Reference	Cell Source	Goal	Bioreactor Setup	Outcomes
Eng et al. [85].	hESCs and hiPSC.	Cardiac organoid differentiation/maturation.	Custom bioreactor with carbon rods in PDMS microchambers.	An increase in frequency leads to increased cardiac gene expression and better beating adaptation.
Yoshida et al. [86].	hiPSC-CMs.	Cardiac organoid maturation.	Organic carbon electrodes in poly(vinyl) alcohol hydrogel chambers.	ES leads to enhanced cardiac troponin expression and sarcomere formation.
Ma et al. [88].	hiPSCs with GCaMP reporter.	Early induction of cardiomyocyte differentiation.	IonOptix C-Pace system.	Organoid contractions were seen after 2 days with ES compared to 7 days without ES.
Zhang et al. [89].	Primary mouse cortical neurons.	Enhanced Neural Organoid.	3D interdigitated electrodes coated with polypyrrole.	ES of organoids leads to enhanced neurite outgrowth and synaptogenesis and rescues the phenotype of NRG1-KO neurons.
Sefton et al. [90].	Primary neural precursor cells.	Formation and differentiation of neural organoid.	Agarose salt bridge stimulation.	ES increases survival and spheroid size. It also increases neuronal expression while reducing astrocyte expression.
Ahadian et al. [92].	mESCs.	Cardiac organoid differentiation.	IonOptix C-Pace system.	ES enhanced cardiac gene expression and the beating area of organoids.

## 6. Discussion/Future Perspectives

Bioreactors have become increasingly valuable for the in vitro culture of organoids. Compared to static culture, bioreactors are more able to support the growth, differentiation, and maturation of organoids for many tissue types. In addition, advanced bioreactor technologies have begun to solve many of the problems inherent to organoid cultures, such as the sufficient maintenance of oxygen and nutrient availability for larger 3D tissue constructs. The primary purpose of most of the bioreactors discussed, namely, stirred, microfluidic, and rotating wall vessel bioreactors, is to provide the enhanced mixing and mass transport needed for larger organoids and other tissue constructs. The methods for achieving this mass transfer differ significantly between bioreactor technologies, and each bioreactor technology has its strengths and weaknesses. The simplicity offered by stirred bioreactors (SBRs) lowers barriers to access, inviting research and researchers from diverse fields of study. However, this simplicity may prevent some users from introducing modifications (e.g., continuous perfusion and the localized application of reagents/biomolecules). The level of control offered by microfluidic bioreactors (MFBs) allows users to precisely regulate fluid flow and composition (e.g., reagents, growth factors); however, challenges such as high fabrication costs, the complexity of operation, and maintaining sterility hinder their broad applicability in organoid cultures. Similarly, rotating wall vessels (RWVs) also allow for fluid flow regulation and have the additional benefit of providing a modeled microgravity environment but suffer from some of the same challenges as MFBs.

Electrical stimulation is a bit of an outlier here, as many ES setups do not provide any media mixing or fluid flow. The primary objective of most of these platforms is “just” to provide electrical stimulation. However, this brings to mind a facet of bioreactor technology for organoids which we find lacking: combination bioreactors. It is difficult to find any examples of bioreactor systems designed to provide multiple types of stimulation in a method appropriate for organoid culture. Indeed, there are some bioreactors, such as a bioreactor designed for electrical stimulation and interstitial flow through a 2D scaffold [97]. This bioreactor, like many others, is not designed or optimized for spheroidal organoid culture. Future research into bioreactors combining multiple technologies, such as microfluidic devices with embedded electrodes for stimulation and electrical monitoring, could be helpful for organoid culture. Another notable facet of bioreactor technology that we find lacking is the absence of bioreactors designed for the mechanical compression/tension stimulation of organoids. One possible reason for this may be that most organoid cultures occur in suspension. However, while many in vitro tissue constructs and cells have been shown to benefit from mechanical stimulation [99], this has typically been performed on 2D or elementary 3D constructs lacking the structure and function required to fit our definition of an organoid, as stated earlier.

We have discussed both commercial and custom-made versions of a variety of different bioreactors. It is important to note that although custom-made bioreactors often represent a significantly lower cost alternative to commercially sold systems, they come with their own challenges. These challenges begin with the construction of the bioreactors, specifically with respect to material selection. When preparing materials for direct contact with cell cultures, it is important to ensure that the materials are cytocompatible and suitable for long-term exposure to cell culture conditions [100]. While commercial bioreactors may take advantage of materials such as polystyrene, polypropylene, polycarbonate, and glass, these materials represent a significant manufacturing challenge in a laboratory setting. Instead, most custom-built bioreactors are made from materials such as acrylic, PDMS, and 3D-printed materials that are relatively easy to fabricate. However, the sterilization of these custom bioreactors can be a challenge. While commercially sold equipment is often either single-use and sold in sterile packaging or designed to be autoclavable, bioreactors made in the lab must be sterilized via other means. While the glass and PDMS often used for microfluidic bioreactors may be autoclaved, acrylics and 3D-printed materials must be sterilized with protocols utilizing ethanol, sodium hydroxide, or microwaves for sterilization, as has been described elsewhere [68,101,102]. Care must be taken when sterilizing the input/output ports and connectors, as these are often a source of contamination, especially in long-term cultures. Despite these difficulties, custom-made bioreactors often represent a low-cost and more accessible alternative to more expensive commercial solutions. Due their flexibility in design, they provide greater control and customization compared to commercial bioreactors.

One area we expect to see significant future development is the incorporation of more advanced genetic engineering approaches into organoid culture. While there are certainly some examples of genetically encoded fluorescent markers used in organoid development (as discussed earlier), their use is not yet universal. Given the many advantages of genetically encoded markers, especially in monitoring changes in gene expression, and the ease with which new genetic tools allow such cell lines to be generated, the use of these tools should become much more widespread in the coming years. Similarly, powerful tools such as single-cell RNAseq that allow for a much more comprehensive measurement of gene expression are rapidly falling in price per sample; thus, they will likely become widely used for the analysis of organoid cultures [103].

While this review has discussed many bioreactor technologies that are currently being used to generate and culture organoids, it is important to discuss a few of the potential applications for these organoids. In theory, every technological advancement discussed in this review brings the field closer to the successful implementation of organoid technology for uses in transplantation, drug discovery, and personalized medicine for the treatment of various conditions [104]. Technologies that allow for organoids to be made faster, cheaper, and more able to recapitulate native, healthy, and diseased tissues will help facilitate the implementation of organoids in a clinical setting [105]. However, utilizing organoids in this way brings many significant technical challenges. Any new organoid system must be validated to show that it is comparable to a native system and able to react in a comparable manner. Differences in iPSCs from different patients may lead to variations in the quality of patient-specific organoids [106]. When using iPSC-derived organoids for implantation, any undifferentiated stem cells still present a risk of tumor or teratoma formation [107]. For these reasons and more, bioreactors represent powerful tools that may be used to generate organoids that are more consistent batch to batch and more defined in terms of their differentiation and malleability upon implantation in vivo, thus leading to better translational and clinical outcomes.

In summary, a variety of bioreactors have been used to improve many aspects of organoid culture, including the survival, differentiation, and maturation of tissue-specific organoids and the maintenance and expansion of pluripotent stem cells. Using bioreactors to finely control and manipulate the in vitro environment is crucial for the successful culture of organoids.

## Figures and Tables

**Figure 1 ijms-24-11427-f001:**
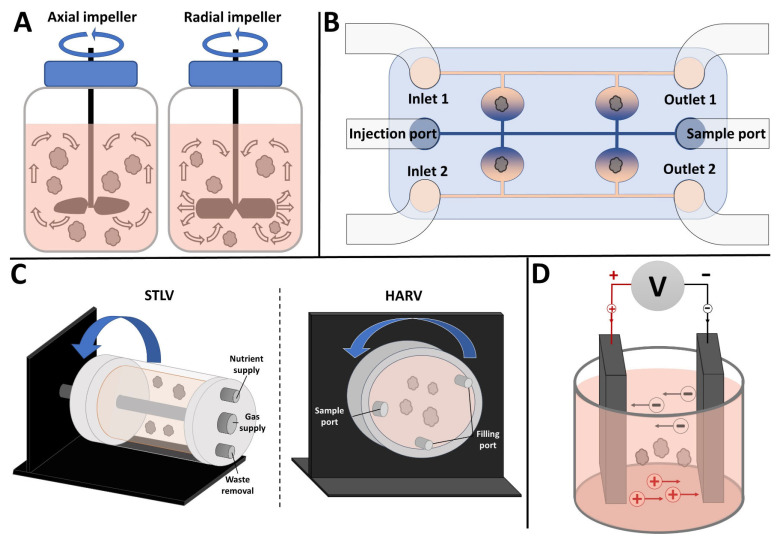
Commonly used bioreactors for organoid culture. (**A**) Stirred bioreactor with axial impeller (left) and radial impeller (right). Arrows show direction of fluid movement. (**B**) Microfluidic bioreactor with separate channels for perfusing different media. (**C**) Rotating wall vessel (RWV) bioreactors: (left panel) Slow Turning Lateral Vessel (STLV) and (right panel) High Aspect Ratio Vessel (HARV). Arrows show direction of rotation. (**D**) Electrical stimulation bioreactor showing two parallel plate electrodes. For details, see text.

**Figure 2 ijms-24-11427-f002:**
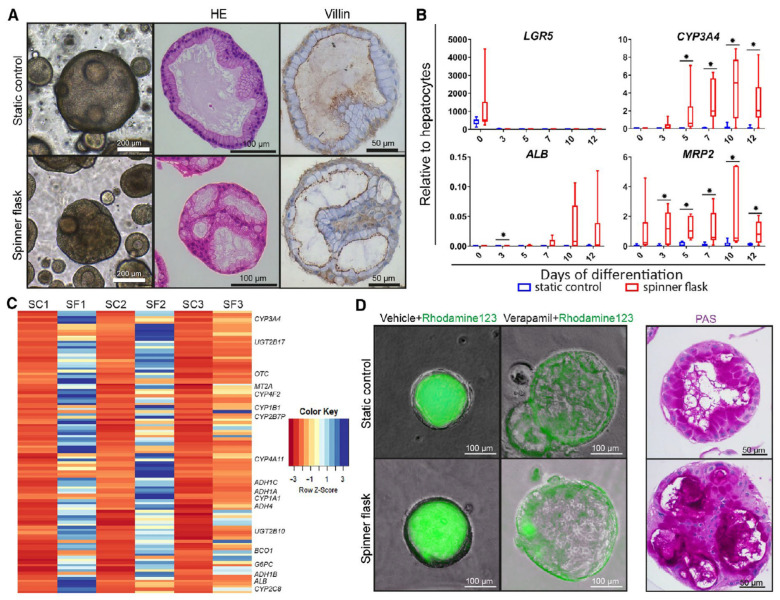
Organoid differentiation into functional hepatocytes in spinner flasks. Organoids were differentiated in spinner flasks or under static controls for 12 days. (**A**) Light microscopy images, HE staining, and immunohistochemical analyses of paraffin-embedded organoids for the canalicular marker Villin-1. Nuclei were counterstained with hematoxylin. Expression of hepatocyte markers in differentiated organoids, determined by quantitative RT-PCR (**B**) and mRNA-sequencing (**C**). (**B**) Transcript levels of *LGR5, CYP3A4, ALB*, and *MRP2*. Graphs indicate five independent experiments for five different donors. Cryopreserved hepatocytes were used as positive control. ** p* < 0.05 (**C**) mRNA sequencing on organoids from three independent donors at day 12 of differentiation. Genes that were more than four-fold upregulated in the spinner flasks at day 12 of differentiation compared with the respective static controls, as shown in the heatmap. Some well-known hepatic genes are annotated. (**D**) Rh123 transport was determined as readout for MDR1 activity, and PAS staining indicated glycogen storage in organoid cells. For details, see text. Adapted with permission from Schneeberger et al. [31].

**Figure 3 ijms-24-11427-f003:**
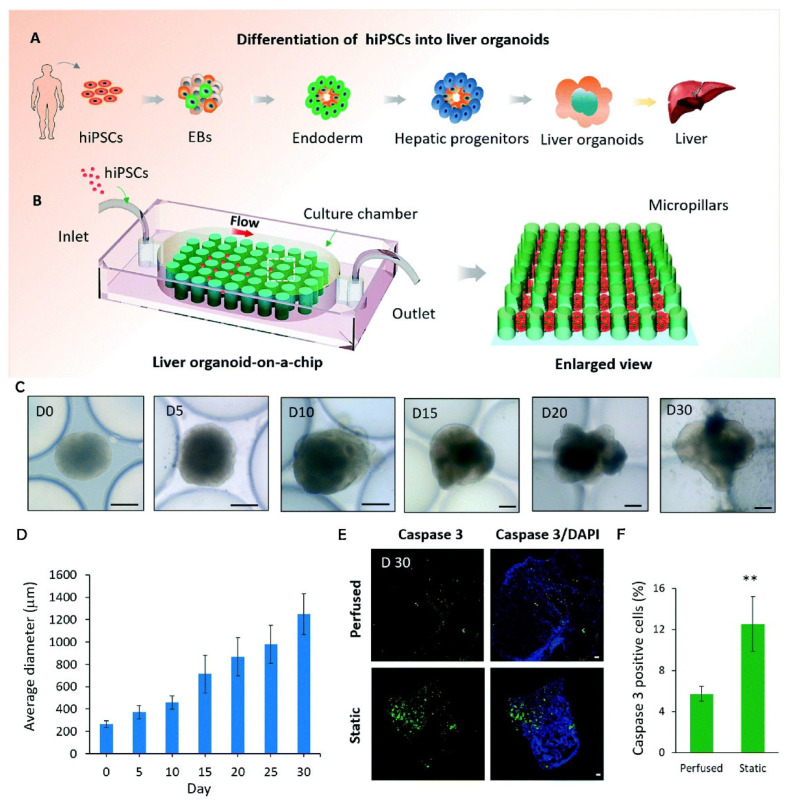
Schematic diagram of the *in situ* generation of liver organoids from hiPSCs using a simple 3D perfusable chip system and the characterization of hiPSC differentiation into liver organoids in the chip system. (**A**) The differentiation process of hiPSC-derived liver organoids in vitro. (**B**) The configuration of the liver organoid-on-a-chip system. (**C**) Representative microscopic images of the spheroids were obtained at different stages of liver organoid generation. Scale bars: 200 μm. (**D**) The size distribution was analyzed using the average diameter of the liver organoids generated on a chip. (**E**) Immunostaining of frozen sections for active caspase 3 to indicate apoptotic cells (green) within the liver organoids at day 30 under perfused or static culture conditions. Scale bars: 50 μm. (**F**) Quantitative analysis of the percentage of caspase 3 positive cells in the organoids displayed from at least five distinct tissues. ** *p* < 0.01 using two-tailed Student’s *t*-test. Reproduced with permission from Wang et al. [44].

**Figure 4 ijms-24-11427-f004:**
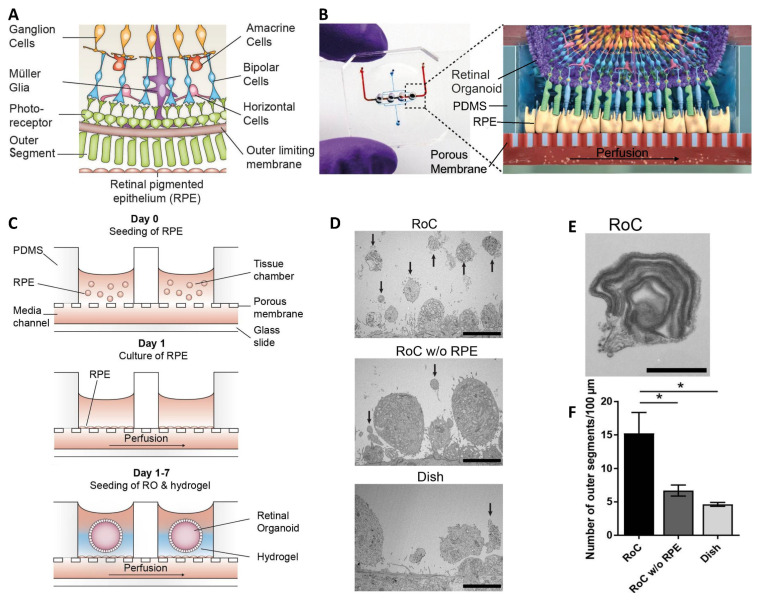
Retina-on-a-Chip. (**A**) Schematic representation of the human retinal composition and cell types in vivo. RPE, retinal pigment epithelium; (**B**) Photo (left) of the RoC (Retina-on-a-Chip) and (right) artist’s rendition of the RO (retinal organoid) photoreceptor and RPE interaction in the RoC. (**C**) (top) RPE cells are seeded into the device, (middle) forming a densely packed monolayer after 24 hr of culture. (bottom) ROs and the hyaluronic acid-based hydrogel are directly loaded from the top into the well and onto the RPE. (**D**) Representative electron microscopic images of the surface of ROs demonstrating outer segment-like structures (arrows) after 7 days of culture (top) in the RoC, (middle) in the RoC without RPE, or (bottom) in a culture dish. (**E**) High magnification image of an outer segment-like structure containing organized membrane disks as seen on an RO cultured for 181 days plus 7 days in the RoC with RPE. (**F**) Number of segments/100 µm RO circumference comparing RoC, RoC without RPE, and dish-cultured RO. ** p* < 0.05. Scale bars indicate (**D**) 5 µm and (**E**) 1 µm. Adapted with permission from Acheberger et al. [45].

**Figure 5 ijms-24-11427-f005:**
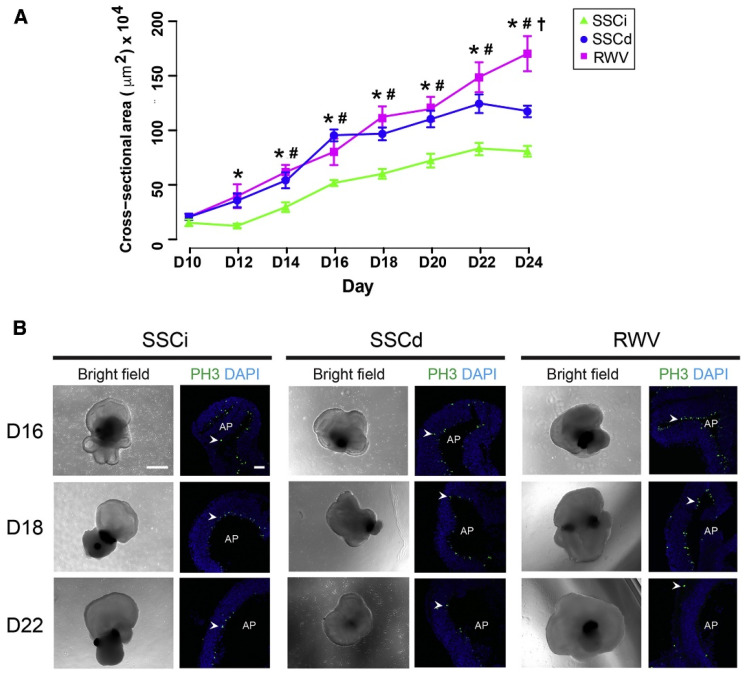
Growth of Organoids Under Different Culturing Conditions. (**A**) Neural retina growth curves throughout differentiation. The largest cross-sectional area of each organoid (in μm^2^) was measured every other day of differentiation. SSCi, intact organoids in static suspension culture; SSCd, dissected neural retina in static suspension culture; RWV, dissected neural retina in rotating-wall vessel bioreactors. The data were obtained from three independent biological experiments (n = 3; three organoids were quantified in each experiment) and are represented as mean ± SEM. ∗ *p* < 0.05 for SSCi versus RWV; # *p* < 0.05 for SSCi versus SSCd; † *p* < 0.05 for SSCd versus RWV. (**B**) Morphology of neural retina (NR) with dividing cells. Phospho-histone H3 (PH3, green) is a marker of proliferating cells. Nuclei were stained with DAPI (blue). Shown are representative figures. Arrowheads indicate relevant immunostaining with PH3. AP shows the apical side of the organoids. Scale bars: 500 μm (left) and 50 μm (right). Reproduced with permission from Distefano et al. (2018) [65].

**Figure 6 ijms-24-11427-f006:**
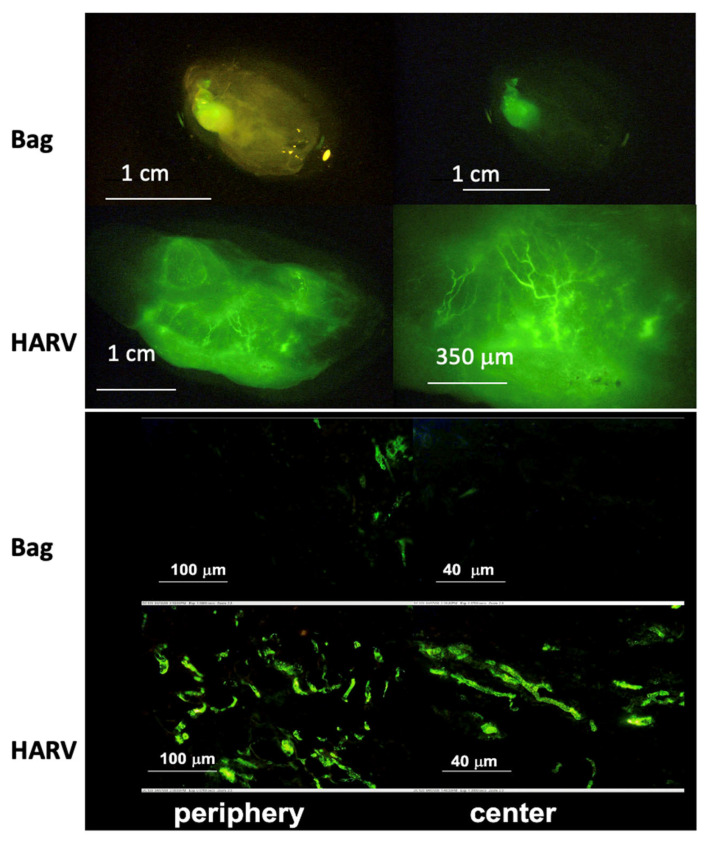
Enhanced angiogenesis in neuroendocrine organoids implanted into a mouse model following dynamic culture in RWV. Bag: static control culture in tissue culture bags. HARV: dynamic culture in HARV-type RWV Bioreactors. (Top) Whole organoid imaging using FITC-conjugated Griffonia simplicifolia lectin (green) to label the vasculature, overlaid with phase contrast (left). HARV condition shows denser vasculature. (Bottom) Fluorescent imaging of sections from Bag and HARV condition organoids showing better-formed vasculature throughout the organoid from the HARV condition. Reproduced with permission from Lelkes et al. [67].

**Figure 7 ijms-24-11427-f007:**
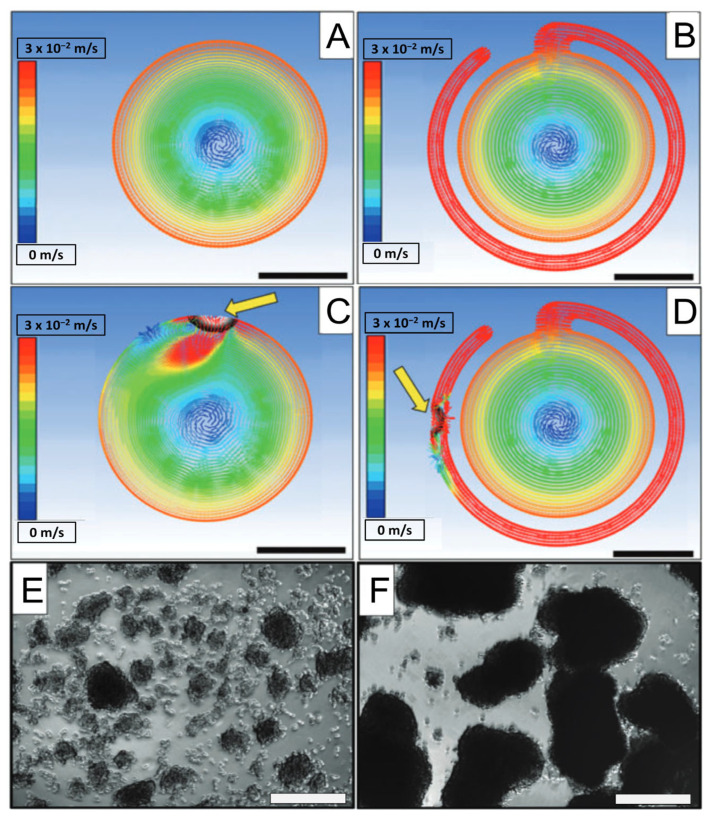
Novel Air Bubble Isolating Bioreactor. (**A**) Computational fluid dynamics (CFD) of traditional HARV design with no bubbles present. (**B**) CFD of bubble-isolating HARV with no bubble present. (**C**) CFD of traditional HARV showing significant disruption in fluid flow due to a bubble being present (indicated by the arrow). (**D**) CFD of bubble-isolating HARV with a bubble present (indicated by the arrow). The bubble is isolated in the channel, leading to no flow disruption. (**E**) Spheroids after formation in the traditional design with a bubble showing small, broken-up spheroids. Scale bars 2.5 cm. (**F**) Spheroids after formation in the bubble-isolating design with a bubble present showing large, well-formed spheroids. Adapted with permission from Phelan et al. [68].

**Figure 8 ijms-24-11427-f008:**
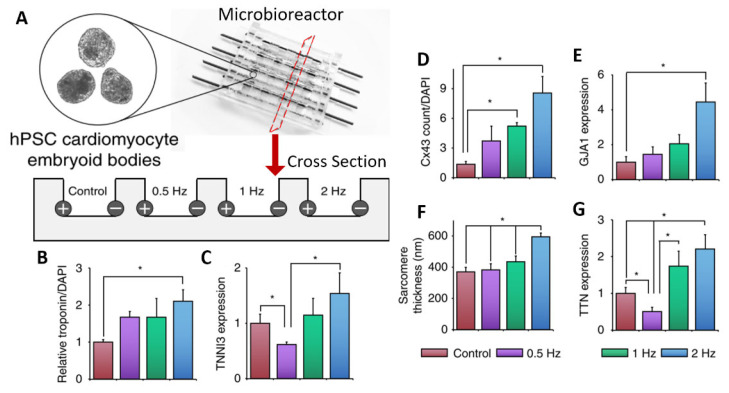
PDMS microbioreactor for electrical stimulation. (**A**) Schematic of microbioreactor set-up showing carbon rod electrodes embedded in PDMS. A cross-sectional view shows electrodes at the edges of PDMS wells with room for organoids in the center. Cells were subject to ES for 7 days at specified frequencies. (**B**–**G**) Quantification of various markers for cardiac differentiation and maturation measured by immunostaining for Troponin T (**B**), Connexin43 (**D**), sarcomere count (**F**), qPCR for Troponin I (**C**), Gap Junction Protein Alpha 1 (**E**), or Titin (**G**). All error bars show the standard error of the mean (SEM), * *p* < 0.05. Adapted with permission from Eng et al. [85].

**Figure 9 ijms-24-11427-f009:**
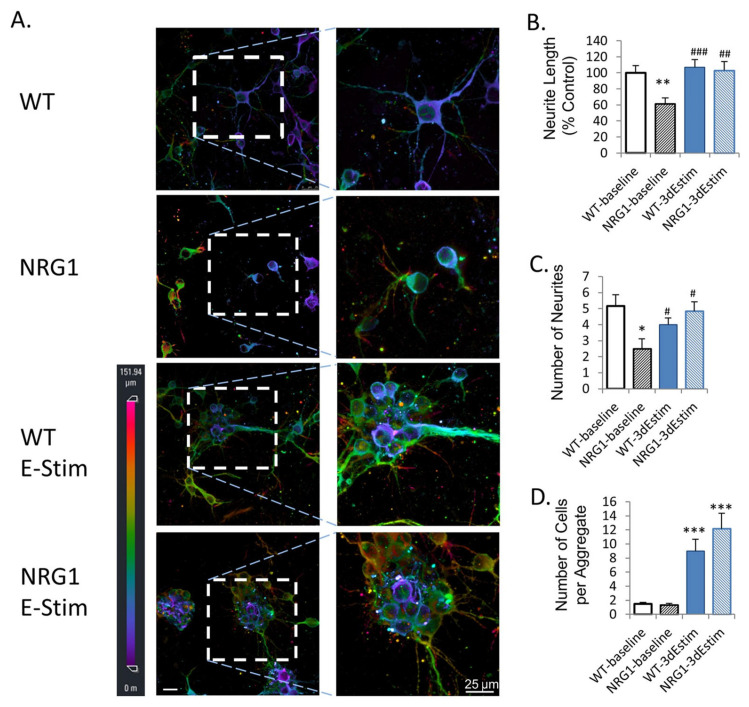
Effects of ES on primary neural organoids. (**A**) MAP2 immunofluorescence imaged using 3D confocal imaging. Conditions are WT (wild type), NRG1 (neuregulin 1 knock-out), WT ES, and NRG1 with ES. All cells are primary mouse prefrontal cortex neurons. (**B**) Quantified neurite length. (**C**) Number of neurons. (**D**) Number of cells per aggregate. * *p* < 0.05 vs. wild-type, baseline; ** *p* < 0.01 vs. wild-type, baseline; *** *p* < 0.001 vs. wild-type, baseline; # *p* < 0.05 vs. NRG1-KO, baseline; ## *p* < 0.01 vs. NRG1-KO, baseline; ### *p* < 0.001 vs. NRG1-KO, baseline. Error bars indicate SEM. Reproduced with permission from Zhang et al. [89].

## Data Availability

Data sharing not applicable.

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
