# Peer review of "Bioreactor Technologies for Enhanced Organoid Culture"

_ijms, 2023, doi:10.3390/ijms241411427_

Round 1
Reviewer 1 Report
Dear Authors,
I studied your manuscript entitled "Bioreactor Technologies for Enhanced Organoid Culture". Although the approach is interesting, some spaces need to be improved in terms of journal quality. In my opinion, the technical significance of the work is still lacking. I recommend a revision before further consideration for publication in the International Journal of Molecular Sciences (IJMS).
1) I do not believe the abstract is adequate. This section requires revision to more accurately reflect the scientific and technical facets of the topic.
2) The weakest component of your manuscript is the small number of illustrative and selected figures. Thus, I suggest you identify some more figures from the cited references that would be representative and include them in the manuscript after receiving proper permission.
3) You mention that electrical stimulation bioreactors have emerged as a promising method for modulating the growth, differentiation, and maturation of excitable tissues. Can you provide some examples of how electrical stimulation has been used to influence the development of specific types of organoids?
4) How does organoid culture in bioreactors impact ethical considerations, particularly concerning the use of human cells? Is there a way to address these issues in future bioreactor research and development?
5) Is there a way to integrate bioreactors for organoid culture with other emerging technologies such as gene editing or 3D printing to make tissue engineering even more advanced?
6) What are some of the primary technical issues that arise when using bioreactors for the culture of organoids, such as maintaining the proper quantities of nutrients and oxygen or avoiding contamination?
7) How may single-cell sequencing or machine learning be combined with the culturing of organoids in bioreactors to further our understanding of the growth and function of organoids?
8) The lack of debate regarding potential clinical uses for bioreactor-based organoid culture may be a cause for concern. While the essay briefly covers potential applications, a more thorough examination of the difficulties and opportunities involved in implementing organoid research in clinical settings would be beneficial.
9) I would also suggest discussing the cost-effectiveness of bioreactor-based organoid culture in the paper, particularly for researchers in resource-limited settings. The cost and complexity of setting up and maintaining bioreactors limit their accessibility to some research groups, despite their improved control over culture conditions.
Reviewer 2 Report
Bioreactors are pivotal in the development and generation of engineered biological products. They simulate the in vivo microenvironment of tissue growth while also providing various mechanical stimuli and biochemical signals to stem cells to effectively generate transplantable organs or tissues.
Nevertheless, the automation will not improve the inherent characteristics of the model on its own; thus, model optimization should precede transition into automation to achieve optimal results. Furthermore, thanks to advances in microscopy, it is also possible to image organoids with unprecedented resolution to gain insight into physiological processes. Yet, challenges remain, including improvements to culture conditions and co-cultures of organoids with other cell types to better mimic native microenvironment and cellular interactions that could potentially accelerate the speed of organoid maturation.
I therefore believe that the authors have paid too little attention to the complex structures for controlling fluid flow and generating the appropriate conditions.
Information on materials for creating bioreactors, including connectors etc., should also have been included.
Information on cleaning and sterilisation of bioreactors was also not included.
Despite these minor shortcomings, I think the work is interesting and should be published.
Round 2
Reviewer 1 Report
Dear Authors,
I would like to express my appreciation for taking my feedback into account and revising your work. Based on the improvements you have made, I am pleased to recommend the publication of your article in its current form.
Thank you for your hard work and dedication to this work.